# Exploring Psychologists' Interventions for Transgenerational Trauma in South Africa's Born Free Generation

**Amy Thandeka Crankshaw and Veronica Melody Dwarika \***

Department of Educational Psychology, University of Johannesburg, Johannesburg 2092, South Africa
\* Correspondence: veronicam@uj.ac.za

**Abstract:** An acknowledgement that the legacy of apartheid lives on in the minds of South Africa's born free generation necessitates an exploration of psychologists' interventions for transgenerational trauma. This research aimed to contribute to research on this subject by interviewing South African psychologists with the ultimate objective of assisting professionals who formulate interventions. Firstly, the ways in which psychologists identify transgenerational trauma were explored. This provided a foundation for exploring the psychologists' interventions for transgenerational trauma and contributed to a discussion of how interventions could be enhanced. Thematic analysis of the semistructured interviews revealed that stuckness paired with guilt, grief resulting from silence and certain manifestations of identity and relationship issues are identifiers of transgenerational trauma. The findings also pointed to the utility of certain approaches to individual, group, family and community interventions. Recommendations for enhancing psychologists' interventions for transgenerational trauma in Gauteng's born free generation revealed the imperative for psychologists to actively engage in professional and personal growth, predicated on the complexity of the challenges within.

**Keywords:** transgenerational; intergenerational; trauma; born free generation; South Africa; psychologists





## 1. Introduction

South African society has an extensive history of race-based oppression [1]. The introduction of formal apartheid in 1948 was preceded by colonial oppression beginning in 1652 [1,2]. Despite the celebrated end of apartheid and the rise of democracy in 1994, South Africa's legacy of trauma prevails in the psyche of its children [1–3]. There is no doubt that traumatic memories live on in the victims of gross human rights abuses, but there is also a need to recognize and treat the trauma transmitted to their descendants. Statistics South Africa [4] defines Generation X adults as those born between 1960 and 1979, millennials between 1980 and 1999 and the born free generation in 1994 and later. The born-free generation has no direct memory of laws enforcing racial classification, official segregation of institutions and interpersonal relationships, passes, the violent repression of resistance or the armed struggle against apartheid [5]. However, they continue to carry the psychological burden of dehumanization and the continued oppression of their forefathers. As a concept, transgenerational trauma (TGT) elucidates the ways in which trauma manifests in subsequent generations and allows for the contemplation of psychological interventions to confront resultant contemporary challenges. Mitigating the damage of TGT requires engagement on multiple societal levels via various disciplines. Given the psychological, cultural, biological, systemic and socioeconomic effects of TGT, solutions require the collaboration of sociologists, historians, anthropologists, politicians, economists and psychologists [6]. This research explores the contributions of psychologists towards consolidating psychological interventions that aim to break the chains of TGT that constrict the minds of the born free generation in one of the nine provinces in South Africa, Gauteng. The province is part of northeastern South Africa and is the smallest South African province.

Transgenerational trauma (TGT) (also referred to as intergenerational trauma) is a term used to describe unresolved trauma that is inherited from previous generations. Scholars such as Yehuda and Lehrner [7], Gottschalk [8], and Danieli et al. [3] advocate the existence of TGT and the role it plays in future conflicts as massive traumas are passed from one generation to the next. It is theorized that when traumatic memories dominate the mental life of victims, behavioral re-enactments emerge at the societal and interpersonal levels [9]. The consequences of large-scale trauma may be a collective mental representation of traumatic events that are re-enacted in the social domain [9]. This phenomenon became an area of academic interest in the 1960s, focusing largely on the children of Holocaust survivors [1,10]. Since then, TGT has been studied in other traumatized groups, including the children of US Vietnam veterans, Australian Vietnam veterans, World War II atomic bomb survivors and pregnant women exposed to the 9/11 terrorist attacks [3,11,12]. Transgenerational trauma (TGT) connected to colonialism, exploitation and discrimination has been studied in African Americans, First Nations peoples in Canada and Australian Aboriginal communities [1,7].

Trauma manifests in first-generation victims in various forms: dissociation (mental, physical or behavioral); self-harm; somatization; interpersonal challenges; and destructive coping mechanisms such as behavioral and substance addictions [13]. Children exposed to parent(s) struggling with complex trauma or PTSD may suffer long-lasting effects from their continual heightened stress responses and may also suffer from affect dysregulation. Dysregulated trauma responses such as hypercortisolism (inability to stop stress response) and hypercortisolism (lack of stress response) result in symptoms such as anxiety and irritability (due to hyperarousal) or flat affect, dissociation and withdrawal (hypoarousal) [14,15]. Impaired parental abilities due to such afflictions can damage the psyche of the child and interrupt their social, psychological, emotional and psychological development [14]. In children whose caregivers have suffered social persecution, this is exacerbated by continued oppression and the transmitted message that their safety is at risk [3,16]. Fossion et al. [17] describe how high levels of emotional and psychosocial disorders such as depression and anxiety (rooted in the grief and terror of traumatic experiences) in the first generation can increase the vulnerability of the second generation to psychological distress and PTSD. Recurring themes in studies of the children of trauma victims are low self-efficacy and inhibition, dependency, interpersonal conflict, uncontrollable anger and frustration, irreconcilable guilt, difficulties with individuation, lower capacity for intimacy, inhibited emotional expression issues related to separation and struggles related to overachievement [14,16–19]. Abrams [16] links secondary traumatization to a living environment where connections and communication patterns have been disrupted.

Gross human rights violations in South Africa share many aspects of the atrocities inflicted on the aforementioned populations, but they are unique in their recency, scale and complexity [20]. The children of apartheid victims and their descendants stand to inherit the debilitating effects of TGT and potentially re-enact the collective mental representation of traumatic events in the social domain. The born free generation is currently 0–27 years old, and in 2018, they made up 45.5% of South Africa's population [4]. Exploring specific psychological interventions for TGT relevant to the South African context is therefore crucial for the mental health of the born free generation and the future of the country.

South African psychologists are tasked with using "any psychological method or psychological counselling" to evaluate, aid and prevent "personality, emotional, cognitive, behavioural and adjustment problems or mental illnesses of individuals or groups of people" [21] (p. 5). The first extensive epidemiological study in South Africa found that 16.5% of adults were afflicted with common mental health disorders, predominately substance abuse, depression or anxiety [22]. This statistic is significantly higher than in other African countries and has been linked to South Africa's high unprecedented levels of economic inequality, poverty, crime and unemployment [22]. In 2019, the estimated population of youths (aged 15–34) in Gauteng was 15.10 million (28.6% of South Africa's total population), and despite being the smallest province geographically, Gauteng contained the largest child population (21% of South Africa's population in 2018) [4,23]. A 2012 study of child and

adolescent mental health in South Africa estimated a prevalence of 17% for psychiatric disorders [24]. This is disturbingly higher than rates reported in other developing countries such as Brazil, Bangladesh and India [24]. Concerning mental health statistics within Gauteng, it was observed in a study of 15 to 19-year-olds conducted in five cities that adolescents in Gauteng reported the highest levels of depression (44.6%), PTSD symptoms (67%) and suicidal ideation (39.6%) [25]. Mental illness in children and young adults is associated with poor academic performance, suicide, substance abuse, unplanned pregnancy and increased risk of psychopathology later in life [24,26]. The etiology of mental illness is complex, but prominent psychosocial stressors include violence, neglect, loss, family conflict, stressful life events and poor social support [26]. Transgenerational trauma (TGT) can be an exacerbating or causal factor in this regard given the aforementioned symptomology and transmission.

Gauteng's trauma-ridden history also makes it an apt site for TGT research. Gauteng was an epicenter for the freedom struggle against apartheid; it is home to the Sharpeville massacre, Soweto Riots and Rivonia Trial that led to heightened police brutality and violent citizen responses [27]. The concept of TGT has utility in exposing how the born free Gauteng population continue to carry the massive traumas of the past and how it is likely to surface "in various symptomatic behavioural forms: suicide, homicide, or other kinds of anti-social behaviour, intrusive memory, psychic paralysis or shutting-down, and various expressions of repetitive interpersonal incapacity" [20] (p. 32). Given the extent of the population vulnerable to TGT and the high rates of mental illness and limited literature on treating TGT within the South African context, an investigation into psychological interventions for TGT is vital for the well-being of Gauteng's born free generation (GBFG).

South African research into TGT is limited with vague recommendations for further studies and therapeutic responses. There is clearly an urgent need to explore how TGT is being recognized and addressed within the South African context and how it could be improved. This research directed such an investigation towards psychologists confronting TGT in GBFG in an attempt to contribute to the local knowledge base of TGT interventions. The research questions were scaffolded to achieve these objectives:

1. How do psychologists identify TGT in GBFG?
2. What interventions are used by psychologists to address TGT in GBFG?
3. How can psychologists' interventions to address TGT in GBFG be enhanced?

In order to understand the interventions selected by psychologists, it is necessary to ascertain how they identify TGT in their young clients. The second research question aimed to identify the psychological interventions that are being used by these professionals. The final research question required a comparison between the data collected from psychologists and the literature reviewed to reveal areas for growth in the treatment of TGT in GBFG.

## 2. Materials and Methods

A constructivist paradigm was used given that the research sought to explore a variety of psychological interventions with diverse underlying theories—there was a need to acknowledge a range of perspectives, appreciating that there is no singular truth [28].

Interviewing psychologists entailed "understand[ing] the complex world of lived experience from the point of view of those who live it" [29] (p. 16). The research questions were addressed through an open-ended inquiry; therefore, the data required interpretation before presenting the tentative conclusions and recommendations regarding psychologists' interventions for TGT in GBFG [28]. Ultimately, this research was aligned with the constructivist notion of ethics, as it aimed to enhance the well-being of individuals in society affected by TGT [29].

A qualitative research design was used in this report, incorporating methods that exemplify the constructivist paradigm, namely semistructured interviews [29]. Given that qualitative studies provide rich descriptions of people's lives and social worlds, it is "useful in understanding the nature, quality, and context of interventions" [30] (p. 255). Qualitative research recognizes the central role of themes and the importance of describing and

interpreting participants' words to construct knowledge, which is essential for assembling and analyzing psychologists' descriptions of TGT and their psychological interventions.

Kaushik and Walsh's [30] description of constructivist qualitative research as "shaped from bottom up" encapsulates the research design of this report (p. 255). First, individual perspectives were considered—relying as much as possible on the participants' perceptions—then, broad patterns were identified, which ultimately led to broad understandings [30].

This study used purposive snowball sampling [31] to obtain its data set due to the niche nature of the research. Finding South African psychologists with knowledge of psychological interventions for TGT for GBFG was potentially challenging if the gap in the literature was indicative of a void in practice.

The inclusion criteria were registration as a psychologist with the HPCSA and experience working with TGT in GBFG. The researcher used networks in the field to identify three psychologists that had knowledge of psychological interventions for TGT, and this formed the initial sample set. The candidates were asked for recommendations from other psychologists who had worked with TGT in GBFG. Exclusion criteria were participants who were not registered with the HPCSA, did not recognize the existence of TGT, had not worked with GBFG or had not considered psychological interventions for TGT. Table 1 contains a summary of the seven psychologists who were interviewed.

**Table 1.** Participants' profiles.

|  | HPCSA Practice Field | Year of HPCSA Registration | Gender | Race | Primary Professional Environment |
| --- | --- | --- | --- | --- | --- |
| Participant 1 | Educational Psychology | 1993 | Female | White | Private Practice |
| Participant 2 | Clinical Psychology | 1993 | Male | White | Tertiary Institution |
| Participant 3 | Clinical Psychology | 2012 | Female | White | Private Practice |
| Participant 4 | Research Psychology | 2001 | Male | Colored | Research Council |
| Participant 5 | Clinical Psychology | 2016 | Female | Black | Private Practice |
| Participant 6 | Art Therapy | 2012 | Female | White | Tertiary Institution |
| Participant 7 | Educational Psychology | 2021 | Female | White | Research |

The study aimed to satisfy the quality criteria for trustworthiness in qualitative research, namely credibility, transferability, dependability, confirmability and reflexivity [32]. To acquire rich data on psychological interventions, the researcher sought to maintain their presence of mind during long interviews with the participant psychologists and planned sufficient time to build rapport and become familiar with the subject matter to be able to adequately assess the information [33]. Several distinct questions guided the semi-structured interviews, and participants were encouraged to elaborate using examples, prompting follow-up questions [32]. Through persistent and careful observation, the researcher identified and detailed the elements most relevant to the research questions [34]. In relation to the issue of transferability, the study endeavored to offer an extensive description of the research process and participant demographics to aid in the assessment of the relevance of the findings to other context [32]. In order to address these aspects of trustworthiness, an audit trail, full transcripts from interviews, process notes and initial coding from the thematic analysis were kept [32]. Confirmability was also enhanced by reflexivity during the research process. Guided by Finlay's [33] reflexivity as introspection approach, the researcher aimed to reflect through self-dialogue by examining own experiences and personal meaning. This was used as a basis for a more generalized understanding and interpretation of the topic. As a middle-class White female instructed predominantly in Western approaches to psychology, a conscious effort was made to consider different cultural interpretations relevant to TGT in the South African context. To aid these reflexivity efforts, a reflective journal was kept throughout data collection and analysis. To address the fairness aspect of authenticity, the researcher observed and negotiated contradictions and competing constructions within the study, intending to present different psychological interventions and perceptions of

TGT [32]. In the pursuit of educative authenticity, the researcher noted any statements from the participating psychologists in the study that indicated growth in their understanding of TGT in GBFG that resulted from interview processes. In terms of ontological authenticity, the researcher documented their own progressive subjectivity and any evidence of this in the participants [32].

## 3. Findings and Discussion

The findings and discussion are organized via relevance to the research questions that guided this research. Themes relating to manifestations of TGT are grouped to address the first research question—demonstrating how South African psychologists identify TGT. Themes indicative of suggested interventions are arranged to address the second research question and grouped in accordance with the mode of intervention (individual, group, family and community). Following the presentation of each theme, the findings are discussed in relation to the literature reviewed. Themes pertaining to how psychologists' interventions can be enhanced are assembled to address the final research question and further discussed in the section outlining implications for practice.

### 3.1. Identifying TGT in GBFG

The symptomatology of TGT within the South African context emerged via various subthemes, which were refined into the following themes: stuckness and guilt; grief and silence; and identity and relationship issues. Another theme that emerged and is reported on in this section is the complexities in identifying TGT.

### 3.1.1. Stuckness and Guilt

The feeling of not being able to move forward in an aspect of a client's life or facet of therapy was linked to guilt stemming from a sense of betraying ancestral narratives. For participant 5, TGT tends to present in clients that are stuck in some aspect of their lives.

> *"The way in which it comes up is in the person feeling stuck so. I'll give an example of that, so you'll find that in most cases there is a Black female, who is quite accomplished and very well educated, but finds that they are quite stuck in their work life and it feels like there's nothing that they can do to get themselves to a different position."*

Participant 5 communicated that this stuckness tends to be rooted in a sense of guilt, stemming from the oppression of their ancestors.

> *"I think that sense of guilt when they talk about the past comes from being the first-generation person to be at university and the first-generation professional, for instance. So the forms of that, in terms of understanding where the person comes from, is being sort of and I guess feeling like you're held back. So even if on the outside everything seems like it's going OK, there's that sense of psychologically and physically feeling like you're being held back by something. And I think a lot of that is because of the processing of traumas that your parents and the people who have gone through before you, what they have not been able to do."*

GBFG also tends to be the first generation to seek psychotherapy; so, they present with the compounded trauma of grandparents and parents who suffered under apartheid.

> *"Generally with people who were from minority groups or Black people whose parents and grandparents experienced apartheid, what happens is they would obviously be the first people in most cases to seek psychotherapy. And you find that a lot of the times what they bring in and what they present is not just their experiences, but experiences that come from their parents. So some of them, I think most of them, it's not even things that they're conscious of. They'll come in because of the depression and will come in because there's a certain area where they feel like I said before, they're stuck. What often comes up in that when you go into the history and look at some of the experiences with parents or grandparents is the fact that there was a lot of racial or social-political problems that they've been through."*

For participant 2, stuckness related to TGT is indicative of being unable to move from the affective instinctual response phase to the "doing the cognitive work" phase in therapy. Participant 2 also linked the complexities of hegemonic narratives and shared memories to certain aspects of guilt felt by the born free generation.

*"The story of the memories themselves can become sort of hegemonic in a way. Like this is 'this is our collective story'. The oral histories you know the retelling and retelling, it eventually becomes a dogma if you like. Maybe part of the work is the complexity of multiple stories. It's not that everybody who is Black has the same story. And maybe your parents had a variation of that story and maybe you didn't have it as bad and you feel guilty about this."*

In addition, participant 5 observed a sense of betrayal and shame when clients' lives deviate from the shared historical experiences of their ancestors, which can result in self-sabotage.

*"In some of the cases that I've worked with, there is a sense of shame for being successful. I've worked a lot with that with first-generation breadwinners. They're doing exceptionally well in most areas of their lives but come to present with signs or even circumstances where they self-sabotage. In cases like that, you have to wonder if this person is trying to unconsciously protect themselves in terms of the intergenerational transmission of trauma. There's a script, a sort of a psychological script that this person has written into the history that they need to pass on. Maybe a script that you can't be too successful. You can't outshine the people that have gone before you, because in a sense, is that betrayal of their suffering and what they've been through. So I think sometimes intergenerational trauma can present in that way."*

Miyoshi's [35] comments on "unconscious conflicts of loyalty" mirror participant explanations for the link between stuckness and the guilt of betraying ancestral narratives (p. 15). Thomas and Bellefeuille [36] explicitly mention "stuckness" in relation to inherited trauma but link this to self-criticism and self-doubt rather than guilt. Authors such as Sotero [37], Hardtmann [38], Möhler et al. [39] and Velde [40] included guilt in their descriptions of TGT symptomatology but did not make explicit links to stuckness or betrayal. Coetzer [41] drew on the works of Ancharoff et al. [42] to describe guilt as rooted in identification with parental experiences of trauma, resulting in the development of parallel symptomatology in the child. Fonagy [43] described guilt and shame as inherited symptoms transferred through disorganized attachment systems.

This theme bears similarities to the literature reviewed but also offers unique insights into how TGT may manifest in GBFG. Guilt was paired with stuckness and linked to a sense of betraying ancestral narratives. This is a valuable observation for psychologists working with GBFG who identify their clients as being unable to move forward in a certain facet of their life. Psychologists who have an understanding of this element of TGT might be better equipped to work with feelings of betrayal, guilt and stuckness stemming from historical trauma.

### 3.1.2. Grief and Silence

Some participants alluded to the inheritance of unresolved grief observable in TGT clients. This transmission seems to be perpetuated through unspoken truths and a disconnect within familial relationships. Participant 4 commented on the harm that comes from first-generation victims trying to protect the next generation by not sharing their traumatic experiences.

*"These things are not spoken about, and the assumption is that by not talking about it you are protecting the children from having to deal with that. But the misconception is that just because you might not speak about something, it doesn't mean that it doesn't manifest in your life. Those are things that affect the children, affect your marriage. So in the end, while the intention of silence is good, it probably does more harm than anything else."*

This ties in with participant 6's observations of the inheritance of unresolved grief through TGT, paired with parental experiences of PTSD, violence, horror and helplessness.

*"A lot of unspoken grief I feel with a lot of transgenerational intergenerational trauma. You're dealing with trauma and PTSD and violence and helplessness and horror all those things. But often you're also dealing with sort of complicated bereavement or disenfranchised grief, so the grief that you weren't able to speak or talk about. It wasn't recognized or validated and you just had to get on with it through development."*

Participant 2 also acknowledged the link between sadness and silence. Feelings of emptiness and heaviness in GBFG could be rooted in the unspoken and unrecognized trauma of first-generation victims.

*"In subtle ways, elements of the trauma get expressed into the next generation through the deficits that came from the original trauma. So that the next generation would feel perhaps the certain sadness or absences of some things that hadn't been explained to them or things that hadn't been done to them. They would sort of be aware of some kind of heaviness which might then affect the way that they live their lives."*

Coetzer [41] also noted that traumatized caregivers' silence tends to enhance rather than allay distress in their offspring, commenting on the hindrance of emotional expression and shared understanding. Coetzer [41] drew on the works of Ancharoff et al. [42] to engage with working models of silence, warning that both silence and overdisclosure can result in identification and re-enactment in the offspring of trauma victims. When Coetzer [41] drew on the works of Bernstein [44] and Velde [40] to describe parental behavior and the perpetuation of TGT, silence about traumatic experiences and symptoms of grief and loss were included. Sotero [37] and Stepakoff [45] also mentioned unresolved grief inherited from forefathers as characteristic of TGT. Duran et al. [46] acknowledged the grief-related affects related to TGT and advocated open communication within families to initiate the healing of ancestral wounds. Similar to the sentiments of the participants in this research, a link was made between grief and silence. Fossion et al. [17] also made this link, commenting on the healing power of Holocaust victims sharing their traumatic experiences with their children. Silence-breaking is described as a powerful way of initiating communal mourning, which implies that manifestations of silence and unprocessed grief in TGT are inextricably linked.

The pairing of grief and silence is a useful insight for South African psychologists in terms of identifying and treating TGT. Clients exhibiting grief without personal experiences of loss could benefit from therapists exploring any ancestral traumas and unresolved grief. It is also useful for psychologists to note that this grief may be characterized by feelings of heaviness, absence and sadness.

### 3.1.3. Identity and Relationship Issues

Transgenerational trauma (TGT) manifesting in clients' relationships emerged as a prominent theme informing how psychologists identify TGT in clients. Participant 2 emphasized the importance of this facet of TGT in the South African context and the potential harm to familial structures.

*"The other area that might come out for me would be around relationships. The idea of what the apartheid system did to the idea of the Black family. It was in a sense a destroyer, affecting people's capacity or ability to have healthy, safe, familiar, positive, intimate relations and setups. So that can end up in situations where in single-parent households there's a disdain or fear of men and masculinity. Men who have had their masculinity thwarted by apartheid enacting that in terms of their own partners and children."*

The violent oppression of Black men, in particular, was recognized by both male interviewees as the source of child–parent conflict. Participant 2 observed fathers "becoming excessively disciplinarian or focusing on the child's success"; in the same way, participant 4 observed this in a workshop between veterans and their sons. Participant 4 shared the

story of an ex-commander from the armed struggle (against apartheid rule) and the effect of this trauma on his relationship with his son. During the workshop, the father and son shared their respective stories with each other, which revealed an unequivocal link between the father's experiences during apartheid and their transmission to his son.

> *"The father was 13 years old in Kagiso. He had his own unit that he was commanding when he had been a soldier. He was basically a child soldier and that was the only thing he knew and then post 1994, he was absorbed into government. But he was strict with his son. In his story, the one thing his son said was that I wish my father can stop being my commander and be my father because he's treating me like I'm someone that is a member of his unit."*

Participant 2 traced issues with intimacy to the breakdown of familial relations during apartheid. The social and economic oppression of certain populations during apartheid led to financial stability rather than emotional intimacy being the foundation of certain family structures. These patterns re-emerged in subsequent relationships, propagating the effects of TGT.

> *"Making it in life in terms of material things and having a stable life, but not that area of personal, intimate mirroring and acknowledgement and letting feelings be expressed because I think the parental generation didn't have the luxury of being able to do that for the next generation. Often, they can experience their parents as loving them but not being able to hold them emotionally and that could then go into the next generation of struggling with having intimate relations because to be able to be intimate with another person you have to be able to be vulnerable and acknowledge your own feelings. If it hasn't been done for you by your parents and you might find that you're struggling in the relationship sphere."*

For participant 3, a fractured sense of self resulting from TGT has dire ramifications for relationships and identity. Participant 3 described this manifesting as "co-dependent behavior where it's all about the other" or defending against a sense of unworthiness and insignificance "through grandiosity or by becoming the center . . . a more narcissistic presentation".

Beyond the familial context, fraught relations in the workplace or school context were suggested as potential indicators of TGT. Instances of self-doubt and identity confusion seem to have inextricable links to past traumas. In participant 1's supervision of counsellors working in schools, TGT manifested in race-based bullying.

> *"They're very, very aware of racial dynamics that are going on between children in their school's bullying and name-calling. . . . You know communication breakdowns."*

There is an interesting association between this pattern of behavior and a case observed by participant 2 in which a 14-year-old boy struggled to navigate interpersonal relationships and competing facets of his own identity in light of his grandmother's transmitted trauma.

> *"His grandmother said to him she hates White people, because of what they did to him and that she didn't want him to be friends with any White people at school, which was very difficult because the school is about 40% Black and 60% White. So to only have Black friends would be possible, but not always straightforward. He was a bit confused because he realized that he was being caught up in a double bind. He needed to, in a sense, honor his grandmother, but at the same time, his friendships with White kids just felt ordinary and natural. It was unweighted by his own personal history at that point, but he was caught up with the idea of being loyal to her."*

Beyond the immediate complications to his social life, participant 2 foresaw the ramifications of this for his future relationships and sense of identity.

> *"I think what it could set in motion was I suppose questioning himself as 'Am I judging people accurately in terms of who should be my friend?' So maybe self-doubting second-guessing himself. Perhaps it might have made him withdraw from White kids or think about them in ways that made those relationships more conflicted or problematic. You*

*could see that sort of this could become a thing would affect how he negotiated friendship, which is, I think, what might be one of the things that are specific to the South African context in terms of our formations."*

Participant 5′s aforementioned comments on TGT manifesting in professional lives are complemented by participant 2′s observations of TGT affecting identity in the workplace. Participant 2 comments on young Black professionals feeling displaced or othered, unable to articulate the source of their discomfort or gain a sense of belonging. This internal struggle could be triggered by past and present traumas as long a South African workplaces are still fraught with microaggressions and unconscious bias.

A multitude of scholars commented on identity and relationship issues related to TGT. Atkinson et al. [31] and Silove [47] link role confusion and a loss of sense of belonging in subsequent generations to dispossession and deprivation experienced by first-generation victims of trauma. Interestingly, they suggest that identity and role confusion can contribute to a loss of direction and reduced efficacy [31,47]. This speaks to issues of identity, belonging and relations in the workplace, especially when considered in conjunction with South Africa's history of violating non-White citizens' rights to work and self-support [31,48]. Like participant 2, Atkinson et al. [31] and Silove [47] seem to acknowledge that TGT is identifiable by people struggling with competing facets of their identity. Lindt [48] describes the absence of a sense of belonging and self-acceptance in those struggling with TGT as originating in fraught child–parent relationships. Coetzer [41] drew on literature that attributed relationship issues to parental detachment, fear of intimacy, limited emotional engagement and difficulties with social interaction. Coetzer's [41] explanations for crises of identity also resonated with participant responses; he noted that the transmission of trauma shatters fundamental assumptions of the self as worthy, and that self-image is damaged by the internalization of parental splitting.

The interviewees linked difficulties with intimate and professional relationships to modeled familial relationships, crises of identity and a fractured sense of self. South African psychologists working with any kind of trauma are likely to make this connection; however, these symptoms become significant when considered in conjunction with the aforementioned symptomatology. Essentially, a psychologist observing identity and relationship issues as well as stuckness paired with guilt and grief related to silence might consider TGT in their case conceptualization and treatment plan.

### 3.1.4. Complexities in Identifying TGT

The complexity in identifying TGT in clients was highlighted by participants 1, 2 and 5. Transgenerational trauma (TGT) was described as irregular and not always very apparent in the way it presents itself, which poses issues for applying a specific treatment model.

*"It's so complex that . . . often when people think about trauma, they think about something that is fixed within a space of time and an event that happened that resulted in a response; but I think with intergenerational trauma and how it's transmitted, it's something that's constantly ongoing and I don't know if there's ever a point in time where you can sort of say that you've dealt with the trauma, and then you can move on. It's not something you can use a model for. It's something that you're constantly uncovering because I think it's all built into the person's unconscious."*

Participant 5 commented on the implicit nature of TGT and how it conditions clients in unique ways—altering their perspectives on how "they take in and how they should be in the world". Participant 5 suggested that this calls for individualized interventions rather than a treatment model that can be universally applied.

*"It's quite hard to distinguish and or have a sort of treatment model or something that you can use for each individual. A lot of the interventions need to be quite individualized because of the very nuanced ways in which it plays out."*

Participant 2 also emphasized that there is not one story of apartheid or experience, and it is important to acknowledge the existence of multiple stories. Participant 1 commented

on her experience of detecting TGT in certain teenagers and adopted children without particular identifiers, stating "I don't know how I've identified it. I just know that it's probably there".

This theme did not align with the literature reviewed but offers insight into why the South African literature on TGT is sparse. It was proposed that individualized and unique presentations of TGT make it difficult to identify and conceptualize through specific symptomology. This poses an obstacle to designing and implementing a treatment model for TGT. Despite this hindrance, the participants offered various interventions that psychologists could use to address TGT in GBFG.

*3.2. Psychologists' Interventions for TGT*

Themes pertaining to suggested psychological interventions were grouped and aligned with the structure of the literature review. This grouping of data highlights the psychological interventions suggested by psychologists for TGT in GBFG (to address the first research question) and primes the discussion of how interventions can be enhanced by allowing global and local interventions to be compared within these categories.

3.2.1. Individual Psychological Interventions

The individual psychotherapies that participants suggested and discussed in detail were psychodynamic psychotherapy, CBT and art therapy. However, the following interventions were also mentioned: "problem-solving therapy", "narrative therapy", "existential therapy" and "a strength-based approach".

Within the subtheme of individual psychodynamic psychotherapy for TGT, comments on attachment and relational approaches were included. Unsurprisingly, the psychologists trained in psychodynamic therapy offered much of the knowledge in this area and acknowledged their recommendations reflected their theoretical leanings.

However, all the participants made mention of attachment theory at some point during their interview. Attachment theory was recommended for case conceptualization, and insecure attachment was highlighted as an important focus of psychotherapy. In addition to helping older clients deal with early developmental trauma, the psychologists with child and adolescent clients discussed the need to work on complex parental and familial relationships disturbing the client's current home life. A psychologist specializing in early developmental trauma described her approach to working with individuals in the following way.

> *"In my work, then you just look at how has that individual learnt from a young age to defend against the shortcomings, against those overwhelming strong feelings of not connecting to a self. It's a spectrum, but the two extremes of the spectrum that we then see in response to that is, where on the one part you will see people sacrificing themselves, so they become co-dependent, so you see a lot of co-dependent behavior where it's all about the other. Or you see the opposite, where people defend against their own insignificance through grandiosity or by becoming the center."*

A psychologist working with children advised that attachment trauma is not immediately apparent, indicating a need for psychologists to treat TGT later in psychotherapy.

> *"You might be dealing with something badly wrong in their attachment early on and it might have to do with, it usually has to do with intergenerational patterns. You often don't hear about family trauma or attachment trauma until you've had a chance to see a person for quite a few sessions that sometimes never comes up in the first session or two."*

Participant 6 emphasized the need to work with ongoing transmissions of trauma in clients' home lives, rather than confronting attachment as a past trauma that can be overcome.

> *"Most of the kinds of interventions that are created for PTSD look at it as Post-Traumatic Stress Disorder. So basically, safety can be re-established in some way and then you can work with that because the trauma is in the past or it was one instance and it's not ongoing. So threat and safety can be re-established, and the threat is gone. But for people*

*who live either in situations or in the home where there's this very complex relationship with the people who are perpetrating violence against you, and developmentally, these vulnerabilities around attachment and windows of development . . . then the symptoms look different, and the treatment actually needs to be different."*

Relational psychodynamic therapy was recommended due to the nature of TGT and the racial dynamics that underpin TGT in GBFG. This approach to psychotherapy advocates foregrounding the intersubjectivities present in the therapeutic relationship and potentially triggering aspects of the therapist's identity. Treating clients whose TGT is rooted in the atrocities of apartheid requires an understanding of intrapsychic and external replications of trauma, making individual psychodynamic psychotherapy an effective intervention.

*"It's very important to foreground the relationship that you have with your patient or client and also be relational, because depending on your demographics you'll trigger something particular for patients. So whatever it is that they've unconsciously taken in and brought with them as a transmission of trauma, will be triggered by certain aspects of who you are as a therapist and what we call transference. The transference relationship with you is not just necessarily based on their own impulses or in their own thoughts, but it's what you represent at a point in history where the trauma happened, and it needs to be spoken about, particularly in therapies when the therapist is White and the client is Black. So I think in those contexts, in the South African context, you cannot talk about that in therapy. It represents something even though it's been like more than a decade from when that happened and when we moved from apartheid. I think most of the current issues around race and obviously definitely intergenerational, it's finding translation within this person or this individual's life and how it then plays out in some of their psychodynamic and intrapsychic processes, which is often intertwined and there's an interplay between the intrapsychic and how a person feels and thinks about themselves internally, and also the external world and the social-political structures that are right now and that were in the past."*

In general, psychodynamic therapy was recommended for individual psychotherapy for TGT in GBFG because of its focus on childhood development and relations with caregivers. Psychodynamic therapists instinctively search for unarticulated sources of trauma and the availability of safe, secure and present primary caregivers. Diminished parental capacity prompted these psychologists to consider the formation of the psyche of the parent and traumatic experiences of previous generations—core elements of TGT. Treatment plans were designed in response to this TGT, for example, "if you're trying to heal damage to self like early development of trauma damage, that person has to have an individual experience like a reexperience of being reparented in a way".

One of the psychologists specializing in psychodynamic psychotherapy commented on the lengthiness of therapy. For clients preferencing an approach requiring fewer sessions, CBT was suggested. With psychodynamic therapy, "it might take longer to find out where the trauma emanates from" but "if the person wanted a different kind of therapy and to be able to manage symptoms, I could refer them to somebody else who does a shorter kind of CBT therapy". Another participant commented on the merits of Trauma-Focused Cognitive Behavioral Therapy (TF-CBT) and the importance of considering evidence-based treatments when working with TGT. Both psychologists proposed clients suffering from TGT could benefit from psychoeducation, with Participant 6 suggesting "there's just so much psychoeducation that has to happen before you can even really engage in treatment". This participant also suggested art therapy for working with youths grappling with TGT.

*"Creative arts nonverbal ways of working for me are very powerful with young people who experience these sorts of things and it's not just because you're working nonverbally and then putting things into language, you're actually accessing different parts of the brain and body."*

A creative art therapy intervention run by art therapy master's students was offered as an example. This program has a problem-solving focus as well as skills around mentaliza-

tion. Clients benefit from reflective practice, active reflection, mentalizing, self-regulation and co-regulation. Participant 6 recommended pairing evidence-based treatment with creative arts or more embodied and facilitative ways of working with trauma.

Recommendations for CBT and art therapy are less prevalent in global literature, making this contribution particularly valuable. The recommendations for psychodynamic psychotherapy resonate with the suggestions of Kestenberg [49], Faimberg [50], Grubrich-Simitis [51], Levine [52], Fonagy [43], and Prager [20], and Fonagy's [43] recommendations in particular bore similarities to the proposed relational psychodynamic approach. The participants' emphasis on attachment theory for case conceptualization correlates with the work of these international scholars, as well as those cited by Coetzer [41], such as Auerhahn and Laub [53], Hardtmann [38], Velde [40], Lindt [48], Kupelian et al. [54], Ancharoff et al. [42], Fossion et al. [17], Abrams [16] and Duran et al. [46]. Although many of the participants spoke about the healing power of storytelling, narrative therapy was not an explicit focus of their individual interventions. Another approach outlined in the literature review that did not emerge as a prominent theme is Gestalt psychology. This suggests that Nagata's [55] and Andrew et al.'s [56] work with TGT could be further explored within the South African context.

### 3.2.2. Group Psychological Interventions

Even psychologists who engage predominantly in individual psychotherapy suggested group therapy for treating TGT in GBFG. Multiple participants supported group work due to the sheer volume of damage and demand for therapy within the South African context.

One of the educational psychologists suggested group processes that encourage communication between parents and school children. Group work was recommended for TGT, as it allows members to "witness each other, share like-experiences and try to understand the meaning of how they feel about certain things", which mitigates "acting out" traumas and "transgenerational feelings" they are unable to verbalize and process. In participant 2′s diversity work, he found White teachers and students "needed to understand that those Black learners in the school weren't there in a sense de novo, they were representatives of histories and stories and they were carrying that into the school space" and "how you cannot say the past is just finished". Another participant referred to groups they ran in Gauteng that helped people with early trauma, identity issues and learning skills around emotional regulation.

Two participants recommended group workshops based on their experiences working with military veterans. Healing was evident through "the pain being acknowledged and that there are other people that can identify with it". The complexities of race-specific trauma within group interventions are a significant concern for South African psychologists. Participant 2 shared his experiences of facilitating groups in which various experiences of TGT were shared: children and grandchildren of apartheid victims had the opportunity to share personal experiences disentangled from dominant narratives; Indian participants shared stories from forced removals and drew attention to the history of indentured laborers; an Afrikaans participant felt she carried TGT as a result of farm murders; and other White participants grappled with complexities of communicating White guilt. There was an acknowledgement that "there is race and trauma work to be done by Black people only with other Black people" but also "work to be done with groups of people separated by histories because the traumas are inflicted with cross race dynamics and power". Participant 5 also commented on the importance of representation in group therapy.

> "A very important way of being able to work through intergenerational transmission of trauma is to have, in a sense, representation. So if it's racial trauma, Black people can't work through it by themselves, they need other races to be able to work through it and similarly White people I don't think can work on it by themselves. So I think the notion of having a group and actually having a group of mixed people, mixed race, in this country where things can be dialogued quite differently and different positions held and different

*things triggered within people, is a really helpful way to work through some of the painful things that have happened in the past."*

There are parallels between this paper's findings and Tarpey's [57] suggestions for school-based interventions. Group counselling was suggested for children affected by TGT, incorporating psychoeducation and experiential therapy. The importance of collaborating with teachers and parents was also emphasized. Parallels between Stepakoff's [45] work and this paper's findings are also valuable. Elements of group work that were found to be particularly meaningful included emotional, cognitive and verbal processing of traumatic memories; narration of trauma stories; empathic listening; creating a safe contained space for expressing emotions; giving other group members a space to express empathy; and assisting with affective expression (which contributed to a sense of being understood and heard). Interviewees' proposals for group facilitation mirrored those of Thomas and Bellefeuille [36]. A nondirective approach was recommended as an effective way of empowering participants and encouraging storytelling and sharing.

### 3.2.3. Family Psychological Interventions

The need for family therapy was often expressed in conjunction with comments on the transmission of TGT. Participant 3 advocated family therapy in light of TGT being systemic and impossible to locate in one part of a system. Participant 3 differentiated between manifestations of TGT at an individual level and a community level and the destructive potential of families prescribing the issue to one individual, for example, a "problematic adolescent" becoming "the focus of the entire family system".

One participant shared their knowledge and experience of working with the intergenerational transmission of violence in families with research and interventions focusing on parents, children and soon-to-be mothers. Another participant commented on the therapeutic value of a workshop between ex-combatants and their sons which enabled them to share their stories with each other. This speaks to the healing potential of narrative family therapy, creating a space for victims of trauma to break the well-intentioned silences that may be inhibiting certain levels of intimacy with their children.

*"With their permission, we would tell the father's story to the son and the son's story to the father. And these are people that live in the same house but realised they didn't even know each other. . . .While the intention of silence is good, it probably does more harm than anything else. One thing that we found with these veterans that we've worked with is the therapeutic value they found in writing their stories and having them told not only to the general public but also to their families."*

These workshops also exhibited the healing potential of group therapy. In separate groups before the father–son engagement, ex-combatants benefitted from sharing their stories, and their sons were able to relate in terms of their experiences with their father.

In the literature reviewed and participant responses, the need for family therapy was highlighted by conceptualizing TGT as a family issue. Like Abrams [16], Hardtmann [38], Ancharoff [42] and Velde [40], the interviewees emphasized the importance of facilitating open dialogue about secret traumas to combat TGT. Auerhahn and Laub's [53] suggestions for parents and children recording and sharing their experiences are also evident in this paper's findings. Akin to the interviewees' experiences, Lindt [48] observed children feeling blamed for their parents' trauma and deprived of unconditional acceptance. Prager [20] also lamented the disruption of child–parent relationships as they relive and recreate their traumatic experiences in the present. These scholars and the interviewees recommended healing relations through shared understanding, which requires breaking the silence around traumatic memories.

### 3.2.4. Community Psychological Interventions

Some of these family-oriented interventions also function at a community level. The interventions for women experiencing violence in pregnancy and intimate partner violence

were delivered in antenatal clinics, and the aforementioned veterans had the opportunity to share their stories with the general public.

Participant 7 was part of a community initiative that involved group work with children which incorporates experiential and narrative therapy. Her experiences highlighted the importance of flexibility and the complexities of designing and imposing community interventions.

> *"It really showed me the type of intervention that we needed or that we will need for intergenerational trauma . . . in that we went in there believing that we were going to be using storytelling and narratives as a way of starting to unpack experiences of poverty, only for our participants to be 100% closed off from the process altogether because the stories that we were sharing were far too close to the bone . . . and in many ways the presentation that we had was also so far from what their personal lived experiences were and we were overshadowing and excluding a large part of their experience through bringing in our own narratives."*

The intervention evolved with the help of drama therapists and an adjusted facilitation style that empowered participants to direct the sessions.

> *"It was incredible to see the shift that happened when we then started to move our bodies and it started to just shift and change the energy because trauma's so heavy. And then the next day we went into a process where myself and the macro facilitator stepped aside and said tell us your story and we got presented with the most heart-breaking narratives . . . we have to be the ones that just hold space, in such a way that we can listen to the narratives and start to support development by asking what do you need? What will that look like?"*

For participant 7, this experience exemplified the need for psychologists to take a grassroots approach to development—ensuring interventions are community-driven and sustained. The community interventions shared by participants differed vastly from the experiences of international scholars such as Atkinson et al. [31], Silove [47], Tsey et al. [58], Frederico et al. [59], Hazlehurst [60] and Phipps and Thorne [61]. While these scholars designed and implemented model-based interventions, the interviewees debated the feasibility of this approach in the South African context. These findings suggested that South African psychologists need to be flexible and less directive in their approach, ensuring interventions are community driven.

The interviewees' suggestions for family, group and community engagement also resonate with the sentiments of Prager [20] and Benjamin's [62] conceptualization of the "moral third" to aid the transition from aggression to recognition, which speaks to the patterns of intergenerational violence voiced by interviewees. Psychologists grappling with systemic violence and continuous trauma may benefit from Benjamin's insights for fostering societal change. The mechanisms she emphasized may appeal to professionals inclined to a relational approach, working with groups or participating in community interventions. Within these settings, a focus on interconnectedness, empathy and humanity aligns with the South African philosophy of ubuntu [62]. The philosophy of ubuntu is embodied in the understanding that there is an interconnectedness among people, their country, their environments and their spirituality—these are all interrelated [30].

Dialogue needs to be facilitated in a way that enhances identification and acknowledges and witnesses suffering. Group facilitators struggling with their own and participants' racially based shame and despair may benefit from developing a consciousness of the moral third. Recognizing the dynamic interplay of acknowledging the suffering of others and the truth of one's own suffering lends itself to replacing denial and indifference with empathy [62]. Psychologists could benefit from being cognizant that dissociation from pain and shame hinders the reparation of connections that have been lost [62].

Prager [20] also applied psychoanalytic mechanisms of change to societal efforts to break intergenerational cycles of trauma. Like the interviewees, Prager [20] advocated public remembering to disable the hold of traumatic memory in the present. However, he similarly recognized the destructive potential of externalizing conflict (when past traumas

are demarcated from present selves) perpetuating othering, villainizing and victimhood. Detaching oneself from collective narratives and ancestral experiences of trauma can be paralleled with the delicate balance required for remembering but also progressing as a society. This brings us to a discussion of the challenges psychologists face within the South Africa context, based on the findings of this research.

*3.3. Enhancing Psychologists' Interventions for TGT in GBFG*

Suggestions for enhancing interventions were elicited by asking participants what advice they would give psychologists working with TGT in GBFG. Thematic analysis also revealed the following themes that proved useful for answering this research's third research question: the complexities of the South African context and personal and professional growth.

3.3.1. Challenges in the South African Context

Insights such as these equip psychologists to work with the unique nuances of TGT within the South African context. Participants alluded to various aspects of TGT that require South African psychologists to have an in-depth understanding of how the past is being carried and transmitted. Psychologists working with TGT in GBFG need insight into the origins of TGT as well as current factors perpetuating the transmission of trauma across generations. Emergent subthemes that informed this section included victimhood and violence, continuous traumatic stress and divergent experiences of TGT. These factors pose various challenges to clients and psychologists working with TGT in the South African context.

In the South African context, TGT is complicated by the shared and divergent experiences of multiple sources of trauma. For example, participant 2 commented on how apartheid's Immorality Act "joined Black and queer people in complicated ways because we both are sets of relations were regarded immoral and so there's intertwining of certain forms of harm". The subthemes of victimhood and violence, continuous traumatic stress and divergent experiences of TGT emerged as intertwined. Participant 4 introduced the notion of generational victimhood, linking it to TGT in SA.

> *"One thing that comes to mind in terms of other types of symptoms is a deep sense of victimhood . . . the notion of generational victimhood and how we define victims and that is also linked to transgenerational trauma. We should look beyond primary victims, we are preoccupied with this legal conception of what a victim is and with the history that we have, we have the centuries of oppression and exploitation, and all of those things with transgenerational trauma there are other victims. And those are victims that should be acknowledged as well."*

Here, participant 4 suggested White South Africans also carry the burden of apartheid when considered through the lens of historical trauma.

> *"You can make an argument for White people also having been victims of apartheid when you look at it from an intergenerational transgenerational perspective. You have this discourse around White privilege and young White people you know, like yourself, that have never lived under apartheid but in a way also paying the price of what was essentially the sins of their forefathers."*

The impact of victimhood was emphasized by Participant 4 and Participant 3 concerning global patterns of oppression and transference of trauma. Participant 4 brought up the link between "what White people suffered in the concentration camps, women and children dying of malnutrition, what is essentially being regarded as genocide and the emergence of apartheid" and Israel's oppression of Palestinians. Drawing on Gandhi's sentiments of victims becoming perpetrators, participant 4 commented on the regrettable cycle of trauma.

> *"These are people that were traumatised on a massive scale; you would assume that someone or even a group of people that have been subjected to that type of experience*

*would be more sensitive and have more empathy towards other groups, and it's the exact opposite."*

Participant 3 also drew on South Africa's trauma-ridden history to explain the transmission of trauma from one population of victims to the next. Cycles of oppression in South Africa were described as "rooted in a history of conflict and oppression around greed and materialism", beginning way before Afrikaner's apartheid rule with various European settlers and particularly during British colonialism. Participant 4 and Participant 3 acknowledged parallels between the unaddressed trauma that contributed to the apartheid regime and contemporary violence. For participant 3, this necessitated working with TGT on an individual and collective level and moving beyond blame.

*"I think in the healing work there has to be an understanding of that there are more complexities because even in working with the damage itself on an individual level, it is so important when I work with clients that they get to a place where they can move from the blame. I'm blaming my parents, I'm angry at them because I'm so wounded, but I have to make them the enemy to validate my own experience. Sometimes it takes a long time for someone to become aware enough of their internal space to become aware enough in tolerating the painful experiences without projecting them. I think on a transgenerational level in our country, and I can only reflect on our country, I think I see it in a lot of spaces the same thing playing out. That's why I'm saying I think there's a collective here. It's not just an individual thing where nations or races or communities are so wounded, but because they can't heal or they don't have access or they don't know how. So they can't heal that wound, so the pain has to be projected. So we will project the pain onto the abuser or the oppressor but that does not heal. That just swings the process around. It doesn't bring healing to anybody in a sense."*

Participant 1 witnessed these cycles playing out in a school context, where children and grandchildren of victims of apartheid continue to be victims of racism and strike back by bullying perpetrators on social media. On a different track, but within the theme of victimhood and violence, participant 6 suggested the normalization of violence (throughout various generations in South Africa) has led to victimization not being acknowledged.

*"Often patients don't even recognize that they've been a victim of some sort of violence. It just feels normal or that they've been a victim of some sort of coercion. And there's a sadness attached to what's that happened, but there's not really like a realization that I'm a victim of this. It's just normal. It happens to me and it happens to others in my family and it happens to my neighbor and if you go talk to the police about it, nothing is going to happen or they're going to say it's your fault . . . Am I a victim? Like what does that mean? Because what would change? Is there sort of a mental health problem but this is just how it's always been? I can't pinpoint one thing in my life that then led to a mental health issue."*

This links back to participant 6's comments on the complexities of working with "continuous traumatic stress" in the South African context. Psychologists working with TGT grapple with trauma that is ongoing in "communities and contexts where violence is pervasive and normal, you can't establish security and even the state systems that are there to protect you, like the criminal justice system, can't be relied upon". Participant 7 also reiterated that with TGT, the sufferings of the individual and community are intertwined, and in the South African context, continuous traumatic stress and systemic oppression compound the transmission of trauma. When asked about how she identified TGT, participant 7 spoke from a family and community systems orientation; the first indication of TGT is the "exclusion or marginalization or oppression or violence in a systemic or direct way" in the community's past and/or present. Then, if you "zoom in on a micro individual level you can really start to identify the discord and the break within the family, you can already see the patterns". Looking into a client's history may reveal a lot of "insecure attachment" and "difficulties with regulation and your fight or flight trauma responses but those will be repetitive and evident across the generations". Participant 7 shared that being

clinically diagnosed at an individual level may reveal symptoms such as depression and anxiety. However, psychologists need to look at "holistic systems" as "lots of challenges with mental health are a result of intergenerational trauma".

Both participants 7 and 6 referred to Maslow's hierarchy of needs in their description of issues South African psychologists face in community work. For participant 6, psychological interventions are inadequate in situations where clients' basic needs are not being met: "You can't think about self-actualization or managing your mental health when your basic needs are not secured and safe". Participant 7 emphasized the importance of thinking beyond individual psychotherapy and working systemically, advocating for accessibility.

> "We have got to, as psychologists in the country, be fighting on a far bigger level. We have the capacity and the ability to indicate how these systems are so detrimental to an individual and collective system, and its access, but advocacy at the same time."

Participant 6 contrasted the work of South African psychologists with other contexts to demonstrate the challenges of working at a community level.

> "In other places I've worked, there's a social worker, there's a psychiatrist. You work as a multidisciplinary team, and everybody is taking on bits and accessing systems that are there. I find that the work here is much harder on the clinicians if you're working at community level because you have to be the case worker and the social worker or at least know enough about those systems to be able to help your client and also manage expectations of those services and the people who deliver."

Participant 6 acknowledged that managing the expectations of patients and the ethical burden of working in under-resourced contexts discourage therapists from taking this route. However, she advised that "one has to think systemically, even if one can't work systemically" and described the importance of incorporating the experiences of previous generations into the clinical picture. Participant 4 lamented psychologists' reluctance to "move outside of their particular clinical approach" and commented on the various theoretical leanings of psychology departments at South African universities. Two of the participants felt that systems-oriented training is preferable to a purely psychodynamic approach when it comes to working with TGT. Participant 3 emphasized the need for healing TGT at a communal level but suggested individual healing is still an effective starting point.

> "I don't believe that we can heal trauma on a national or a global level until individuals can heal. Otherwise, if this is a person's experience of self is fragmented they will have a distorted engagement with their emotional experience. They will either project it, suppress it, or be grandiose about it. And those are the kind of patterns that repeat a national or global trauma."

An important acknowledgement by all the participants was that there is not one story or experience of apartheid. The theme of individual vs. collective trauma again comes into play. On the one hand, collective memories and shared experiences can be potentially healing. However, the imposition of collective memory can stifle the individual's need to share their unique traumas and result in identity issues in subsequent generations. Interestingly, the suggestion that healing begins at an individual level is contrary to Prager's [20] proposal that social collectively should bear the burden of TGT to alleviate the burden of individuals suffering on behalf of their ancestors.

### 3.3.2. Personal and Professional Growth

None of the participants felt they had received specific training for TGT but had furthered their knowledge through self-study. Two participants agreed that a module in TGT would be greatly beneficial to psychology students. Another participant suggested training as part of Continuing Professional Development. One psychologist furthered her knowledge through an intergenerational trauma reading group facilitated by the South African Psychoanalytic Association (SAPA). Participant 3 conferred the efficacy of other

groups run by the SAPA which contributed to her personal and professional growth. In small and large group settings, a variety of professionals from different helping professions benefitted from sharing and witnessing each other's traumatic experiences.

> *"We had people from all nationalities, all ages, backgrounds, or socioeconomic statuses participate. It was four or six weekends over a year, running for two years. There were group experiences where people could share in an open forum, so the large groups, and we had small groups in which people shared more personal stuff. In a small group experience of about 12 people with two facilitators, people just shared some of the trauma that they experienced as an individual. But then we also had opportunities where there were large group experiences. Now a large group consisted of about give or take 50. People from all races and ages and genders and sexual orientations, facilitated by about 10 facilitators. It was just an open unstructured format to get people to try and share their experiences and get other people to witness it. Which is quite powerful, so it wasn't a formalized study, but it was an experiential experience in the impact of intergenerational trauma in South Africa."*

An important aspect of personal and professional growth that was reiterated by participants was interrogating one's own positionality. As part of this, participants reflected on and worked through their own inherited traumas. Participant 3 emphasized the importance of healing one's own trauma for psychologists working with TGT and aspiring to consistently grow and learn to become a capable presentable other. For participant 2, understanding the power and role of history is an important element of intellectual growth. Reading was recommended as a way to develop theoretical tools and aid one's own personal work on differences. Participant 2 described understanding your positionality in relations with clients as non-negotiable and emphasized the importance of "retaining a sense of humility and openness to being wrong". Understanding one's own intersectional identity in relation to others might even be brought into individual and group work: "I often start off by telling my own story to say this has positioned me to be here and connect to your stories".

For participant 6, if positionality—such as White privilege—is not carefully considered, psychologists run the risk of "coming in with DSMs and making a diagnosis that actually doesn't fit the context or the patient". Instead, South African psychologists need to consider the impact of social and structural determinants of health on individual health and adopt a less pathologizing view. Participant 6 advises psychologists to see their role as amplifying their client's innate resources through interventions that enhance "a sense of agency and a sense of problem solving from the patient themselves" and "highlight their own resilience and capacity". Giving advice and lecturing are counterproductive for victims of trauma "who have experienced lack of control and helplessness in their lives".

Participant 6 advised that the literature that could aid psychologists' growth in this area is South African scholars' work on continuous traumatic stress. One participant described their engagement with critical race theory, which informed their personal interrogation of positionality and expanded their understanding of tension within their clients.

> *"You're being asked to hold onto ideas about race that are congruent with the narrative and experiences of the older generation which a younger person might, in a more postmodern sense, have felt less connected to."*

Working with TGT in South Africa and enhancing interventions requires engagement with systemic obstacles to healing. Whether a psychologist is working at an individual, group, family or community level, the findings suggest that interventions can be enhanced through personal and professional growth and by considering contextual challenges. These themes are vital for answering the third research question, addressed in the following discussion of implications for professional practice.

## 4. Conclusions

Based on the research findings and literature reviewed, this section proposes practice, research and policy recommendations for psychologists working with TGT in GBFG. For each section, recommendations for identifying TGT and enhancing interventions are included. Prior to these recommendations, it is vital to acknowledge the limitations of this research.

### 4.1. Limitations

The limited scope and small sample size of this research limits the weight of the practice and policy recommendations that follow. A wider scope and larger sample size would have yielded richer and more dependable data; so, recommendations for further studies are proposed in this section. Given this research's strong regional focus, the transferability of the findings to other parts of South Africa is compromised. Replications of this study in various provinces could enhance the dependability of the findings and the credibility of recommendations, especially those with national reach if subsequent studies are consolidated. Although care was taken to counteract researcher bias, the constructivist paradigm underpinning this paper assumes objectivity an impossibility. As such, any replications of this study should contemplate the diversity of researchers and participants and aspire to work reflexively. The lack of previous studies focused on this area of research was also a limiting factor. The aims and objectives of the research may have been better refined had a stronger foundation of South African literature existed. An alternative sampling method may have resulted in a wider range of proposed psychological interventions for TGT. Based on the limitations of this research, it is not possible to make wide recommendations; however, the following suggestions for policy and practice may prove useful if further studies yield similar findings.

### 4.2. Implications for Professional Practice

Since the first research question was formulated to assist professionals seeking to identify TGT in GBFG, it is interesting to summarize the symptomology of TGT present in the literature reviewed that was and was not voiced by interviewees. The following manifestations of TGT were mentioned in the global literature and the interviews: interpersonal conflict; low self-efficacy and inhibition; anger and frustration; irreconcilable guilt; dependency; lower capacity for intimacy; inhibited emotional expression; issues related to separation and individuation; avoidance; repression; shattered fundamental assumptions (of the world as benevolent and meaningful and the self as worthy); feelings of persecution; high levels of emotional and psychosocial disorders such as depression and anxiety; social withdrawal; the impenetrability of bodily and affective signals; narcissistic grandiosity; helplessness; unconscious conflicts of loyalty to traumatized ancestors; stigma and shame; unfounded guilt and absence of sense of belonging; disempowerment; vulnerability; and destructive inaccurate narratives of past traumatic experiences [14,16–19,35,39,40,42,43,45,48,49,63].

Other manifestations that have potential utility for identifying TGT but were not explicitly mentioned by interviewees include nightmares; flashbacks (of the caregiver's traumatic experience); transmission of emotional messages of the fate of relatives (resulting in an emotional void in the child); preoccupation with overachievement; parents assigning their children parental roles of happiness and protection provision; intrusive memories of parents' traumatic experiences; transposition of murdered objects; confusions of identity between victim and torturer; paranoia; racialized stress and discrimination manifesting as restlessness, sleep disorders and muscle tension; heightened sensitivity to power imbalances; distorted cognitions about race, privilege and power dynamics; anxiety provoked by triggering stimuli; and poor academic performance resulting from low self-efficacy [18,39,40,42,49,63,64]. Although psychologists working with TGT in GBFG may find these lists of potential manifestations useful, the findings suggest professionals should remain cognizant of the complexities of identifying TGT. Participant responses

suggested that TGT conditions clients in unique ways and is difficult to conceptualize through specific symptomology.

The findings consolidated to answer the second research question suggest that psychologists can consider individual, group, family and community psychological interventions when treating TGT. Psychologists working with TGT in GBFG may contemplate using the individual psychotherapies that participants suggested (augmented by substantiating literature), such as psychodynamic psychotherapy, relational psychodynamic therapy, CBT and art therapy. Psychologists confronted with TGT in school contexts may find the parallels between this paper's findings and Tarpey's [57] suggestions for school-based interventions useful. Those grappling with the complexities of facilitating groups with divergent experiences of TGT might benefit from the insights of the psychologists in this research, as might family therapists working with GBFG, as well as psychologists involved in community interventions. These professionals might also benefit from considering the approaches outlined in the literature review, especially those seeking theoretical foundations for case conceptualization and treatment plans.

Examining individual, group, family and community interventions illuminated the multilevel dynamics of TGT and the interconnectedness of systems. Psychologists applying ecological systems theory may find utility in the groupings of the literature and findings presented in this paper, as well as Appendices 1–4. Sotero's [37] Conceptual Model of Historical Trauma, Menzies's [65] Intergenerational Trauma Model and Goodman's [66] Transgenerational Trauma and Resilience Genogram may inspire ecosystemic case conceptualizations. Serdarević and Tahirović's [67] Ecological Model and Corresponding Psychological and Psychoeducation/Advocacy Interventions link each level of ecology (individual, microsystem, mesosystem, exosystem and chronosystem) to a TGT stressor and purposeful interventions for each level. Atkinson et al.'s [31] ADAPT Model and the Impact of Threats to each Pillar could help psychologists link individual experiences of TGT to collective traumatic experiences—but also delineate between these experiences. This model suggests normative adaptive responses for each pillar, which may be useful for identifying protective factors and establishing goals for therapy. Applying these models may help psychologists overcome some of the challenges outlined in the findings section and lead to comprehensive case conceptualizations that inform effective interventions. However, it must be emphasized that the findings of this paper suggest a nondirective, community-driven approach, hence the recommendation that these models are useful for conceptualization rather than an inflexible implementation of proposed interventions.

The findings related to the third research question suggest that South African psychologists could benefit from specific training in TGT, but in lieu of this, self-study is paramount. Psychologists working with TGT in GBFG could enhance their interventions by reading extensively on the subject and seeking opportunities to further their knowledge via reading groups, conferences and workshops. Based on participant responses, SAPA is an organization that offers such opportunities for professional growth. This paper's findings suggest that group and individual work are valuable for personal and professional development. Examining one's positionality (through introspection and cross-cultural group engagement) allows therapists to consider intersectionality and how it manifests in therapeutic relationships. Given the racial oppression underpinning South Africa's TGT, South African psychologists need to be cognizant of their identity markers and what this could trigger in their clients. Neglecting to do so has dire implications; for example, this research suggests leaving White privilege unchecked could lead to misdiagnosis and misguided treatment plans for TGT.

It is also vital that psychologists working with TGT in GBFG have an in-depth understanding of challenges unique to the South African context. Extensive knowledge of South African history and how this informs current systemic issues is key to understanding how past trauma is being carried and transmitted. In the South African context, apartheid exemplifies the hurt a traumatized population can inflict; the participants in this research drew attention to the TGT carried by Afrikaners in the wake of British oppression. The pat-

tern of victims becoming perpetrators is equally essential for understanding interpersonal violence in South Africa. Psychologists aspiring to grow professionally could benefit from considering how TGT is characterized by continuous traumatic stress and complicated notions of victimhood due to our history.

Furthermore, the findings suggest psychologists should carefully delineate between collective and individual experiences of TGT. In some cases, a shared history of ancestral trauma aided the psychologist's individual interventions. Healing in family, group and community interventions was often attributed to the expression of shared experiences of TGT. However, in other instances, the internalization of collective memory contributed to psychological distress, and clients benefit from sharing their own divergent experiences of TGT.

Another challenge South African psychologists face is working systemically. The findings correlate with the assumptions of historical trauma theory, indicating that TGT requires psychologists to consider various levels of intervention. However, the implications for practice become complex when considering clients whose basic needs are not being met. Some participants debated the usefulness of psychological interventions in these contexts, but others suggested that working systemically involves advocating access to services. The reluctance of psychologists to work at a community level was attributed to the ethical burden of working in under-resourced contexts without reliable public infrastructure. Multidisciplinary teams and other necessities for systemic interventions are undermined by a shortage of public health professionals and ineffective public protection services. Psychologists may also feel ill-equipped, as not all South African universities' psychology departments focus on training their students to work systemically. In light of these obstacles, the first step towards enhancing professional practice may well be thinking systemically, even if one cannot work systemically.

Psychologists working with TGT in GBFG could enhance their practice by heeding the advice of this research's participants. Personal and professional growth requires reflexive engagement with the subject and interrogation of one's positionality. With limited opportunities for specific training, psychologists are encouraged to read extensively and pursue collaborations with professionals with similar interests. The results of this research suggest that identifying TGT can be problematic, despite common observations of stuckness rooted in guilt, grief resulting from silence and relationship identity crises. Unique presentations and experiences of transmitted trauma limit the applicability of a universal treatment model and task psychologists with designing or adapting interventions. Psychologists working with GBFG could benefit from recognizing the specific symptoms of TGT in their clients and tailoring interventions to combat particular manifestations. The gap in the South African literature necessitates the contemplation of symptomatology identified by international as well as South African scholars, as outlined in the findings above. Contemplating this research's implications for practice revealed the merits and obstacles to working systemically within the South African context. Psychologists working with TGT are urged to think systemically, even if they are unable to overcome systemic challenges. In light of the findings of this paper and the theoretical foundations of historical trauma theory, ecological systems theory is recommended for case conceptualization and treatment planning. The merits of this approach are outlined in the theoretical framework section of this paper and used to substantiate the conceptual structuring of this research. The models included in the appendices of this research could serve as aids to case conceptualization and inspiration for more flexible interventions.

### 4.3. Policy Recommendations

The findings suggest that psychologists working with GBFG lack specific training for TGT and tend to rely on self-study to enhance their expertise. Although introspection and extensive reading aid personal and professional growth, there is a need for institutions to offer courses on addressing TGT. Associations such as the Psychological Society of South Africa (PsySSA) could follow the lead of SAPA and create a space for psychologists to share

their knowledge and experience with TGT. Recommended interventions for TGT could be featured in their webinars, publications and interest groups. Professional bodies such as the HPCSA could encourage Continuous Professional Development programs to include TGT as a focus and ensure working with TGT becomes a core competency for South African psychologists. South African universities and colleges could incorporate modules into their curriculum that better equip psychologists to address TGT.

*4.4. Research Recommendations*

Much of the existing literature on TGT in South Africa relies on the theorizing of international scholars. Except for Adonis's [1] work, the proposed manifestations of TGT seem to be based on studies of foreign populations. As such, empirical research on TGT symptomatology in South Africa would be a valuable contribution to the field. Empirical studies assessing the efficacy of interventions implemented by South African psychologists would also provide a foundation for evidence-based practice for TGT. Given the narrow scope of this research and the small number of participants, replications of this research on a wider scale could yield richer data. Snowball sampling contributed to the majority of participants recommending psychodynamic interventions. The expertise and knowledge of psychologists in various provinces with diverse orientations could be consolidated and used to help psychologists identify TGT in South Africa and enhance their interventions. This research began with the assertion that solutions require the collaboration of sociologists, historians, anthropologists, politicians, economists and psychologists due to the cultural, biological, systemic and socioeconomic effects of TGT [5]. The reviewed literature revealed that there is a need for future academic collaborations to include the research and insights of South African psychologists; such insights may shed light on how collaborations from a multidisciplinary lens could be useful in enhancing multimodal interventions to support individuals and groups with TGT.

**Author Contributions:** Research and original draft preparation, A.T.C.; reviewing, editing and supervision: V.M.D. All authors have read and agreed to the published version of the manuscript.

**Funding:** This research received no external funding.

**Institutional Review Board Statement:** The study was conducted in accordance with the Declaration of Helsinki, and approved by the Institutional Review Board (or Ethics Committee) of the University of Johannesburg (Sem 2-2021-040).

**Informed Consent Statement:** Informed consent was obtained from all subjects involved in the study.

**Data Availability Statement:** The data that support the findings of this study are available from the corresponding author, Veronica Melody Dwarika, upon reasonable request. The data cannot be shared openly to protect study participant privacy.

**Acknowledgments:** We would like to thank Nikki Watkins for English language editing.

**Conflicts of Interest:** The authors declare no conflict of interest.

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
