# Peer review of "Exploring Psychologists’ Interventions for Transgenerational Trauma in South Africa’s Born Free Generation"

_traumacare, doi:10.3390/traumacare3040020_

Round 1

Reviewer 1 Report (New Reviewer)

Exploring Psychologists’ Interventions for Transgenerational Trauma in South Africa’s born-free generation

This is a fine detailed study of 9 South African psychologists’ views on transgenerational Trauma (TGT) in South Africa.  The focus is on how the “born free generation” (post-apartheid) has been affected by their parents’ traumatic experiences with apartheid.  The sample size is small, but the idea behind it is clever: investigate how psychologists dealing with TGT have come to conceptualize the problem and formulate interventions. 

The paper’s strongest attribute is its extensive citation and use of the TGT literature that has emerged from the Holocaust, as well as other venues, for evidently there is virtually no South African published research on the topic.  The author’s knowledge of the TGT literature is extensive.  His/her ability to draw upon this literature to formulate the issues, as well as interpreting psychologists’ responses in terms of this literature is impressive.

My primary objection (which is at the same time a compliment) is that there is just too much literature, too much covered.  For example, the author mentions (PDF, p 2) and cites some work on the epigenetic transmission of trauma, for which there is some evidence.  But the author is utterly unable to study this phenomenon in South Africa because the psychologists are unable, and so it just clutters the argument.  This is hardly the only case.  Prune the literature to focus more precisely on the therapeutic approaches about which his/her informants (the 9 psychologists) are most enthusiastic.  Relational psychotherapy, because attachment seems an important issue.  Group therapy among mixed-race groups, and family therapy, particularly with adolescent “born free” children.  There are one or two others.

I’m struck, and the author is too, by how the psychologists are winging it, having had no instruction in TGT, they themselves seem often without a clear idea of how TGT works.  Guilt at parents' experience leading to children’s “stuckness,” is a common experience of psychologists, about which they are puzzled.  Their puzzlement, though, is itself interesting research finding with policy implications.

In conclusion, it’s a clever research idea, the N is small but this is a preliminary study, and it certainly sets an ambitious research agenda for others to follow, while laying out the literature on TGT that must be dealt with.    What it needs is more focus on fewer key themes raised by the psychologists.  This would allow the literature pruning necessary.  But to repeat myself, this fault is also a virtue, for the study’s comprehensive familiarity with a wide range of TGT research is also its second strongest point.  The strongest is the original idea of interviewing not those who suffer from TGT, but the psychologists who work with them. 

Minor revision and definitely publish. 

Author Response

Dear Reviewer, thank you for the feedback and suggestions for pruning of the literature. Please find attached for how this has been managed in the revision.

Reviewer 2 Report (New Reviewer)

The research explores the contributions of psychologists towards consolidating psychological interventions that aim to break the chains of transgenerational trauma (TGT) that constrict the minds of the born free generation in one of the nine provinces in South Africa. The research questions were mainly concerned with how psychologists identify TGT, what interventions are used by psychologists to address TGT and how psychologists’ intervention to address TGT can be enhanced. The authors rightly pointed out that identifying TGT remains problematic. It is also the authors’ view that solutions to TGT require the collaboration of sociologists, historians, anthropologists, politicians, and economists. The authors should have explained in more depth what the role of politicians, economists and anthropologists would be in this endeavor.  Otherwise, I find this research to be novel and timely, especially given South Africa’s apartheid past.  

Author Response

Dear Reviewer, thank you for the feedback. Please find attached for how this has been managed in the revision.

Round 2

Reviewer 1 Report (New Reviewer)

The revisions meet my objections.  My main objection was excessive citing of sources irrelevant to their argument.  This was corrected, and I believe the paper can now be published as is.   A good job. 

This manuscript is a resubmission of an earlier submission. The following is a list of the peer review reports and author responses from that submission.

Round 1

Reviewer 1 Report

Exceptional food for thought - Well done 

Author Response

REVIEWER 1

Minor spell check needed.

No further revisions required

Spelling of words that contained 's' have been changed to show the 'z' where necessary.

Example cognisant changed to cognizant; recognise changed to recognized; etc. See track changes for all effected spelling changes.

Reviewer 2 Report

A fascinating and substantive study that has been conducted with depth, nuance and attention to the historical and cultural context. 

I would suggest developing the concept of Ubuntu a bit more as it is quite relevant and only mentioned briefly.

Author Response

REVIEWER 2

Minor spell check needed.

I would suggest developing the concept of Ubuntu a bit more as it is quite relevant and only mentioned briefly.

The spelling of words that contained ‘s’ have been changed to show the ‘z’ where necessary.

Example cognisant changed to cognizant; recognise changed to recognized; etc. See track changes for all effected spelling changes.

The philosophy of ubuntu has been expanded (see line 737-738). The authors consider that a more elaborate explication will detract from the key issues elucidated in that section.

Reviewer 3 Report

Transgenerational trauma is an important, relevant topic to many of our patients, and so I was glad to see an exploration of psychologists' interventions for this issue.

The article is well written, but my main disappointment is that it lacks scientific soundness somewhat. I would have liked to have seen some measurements of some features and some tables with data, and then some conclusions about what objectively has been found.

Similarly, some of the writing style, in my opinion, should be changed to better reflect more of an objective scientific article. For example, the opening sentence line 25 "..society remains enchained by its extensive history of race-based oppression..."  I do not debate the author's claim nor diminish its importance, but such language is hard to process scientifically. It would have been better to start with the issue of intergenerational trauma, i.e., try to stick to the scientific.

Also similarly, if you write some numbers, now rather than me passively reading and thinking, 'oh that's bad' my mind is better able to process numbers and objective data. For example, in line 103 there is mentioned that 16.5% of all adults in South Africa were afflicted with common mental health disorders and that it is much higher than other African countries, with the implication it is related to TGT. However, I know offhand, that, for example, in Canada, a relatively quiet country (although certainly not spared from TGT) over 50% of the population by the time they are about 40 years old will have had a mental disorder and the incidence is about 20% per year (let me find some references... Smetanin et al 2011 The life and economic impact of major mental illnesses in Canada). That then takes away credibility from the article.

In summary, the article is reasonably well written and involves a relevant, important topic. I do not disagree with the authors' feelings, but respectfully would advise trying to make the article as objective and scientific as possible.

Author Response

REVIEWER 3

Minor spell check needed.

The article is well written, but my main disappointment is that it lacks scientific soundness somewhat. I would have liked to have seen some measurements of some features and some tables with data, and then some conclusions about what objectively has been found.

Some of the writing style, in my opinion, should be changed to better reflect more of an objective scientific article. For example, the opening sentence line 25 "..society remains enchained by its extensive history of race-based oppression..."  I do not debate the author's claim nor diminish its importance, but such language is hard to process scientifically. It would have been better to start with the issue of intergenerational trauma, i.e., try to stick to the scientific.

Also similarly, if you write some numbers, now rather than me passively reading and thinking, 'oh that's bad' my mind is better able to process numbers and objective data. For example, in line 103 there is mentioned that 16.5% of all adults in South Africa were afflicted with common mental health disorders and that it is much higher than other African countries, with the implication it is related to TGT. However, I know offhand, that, for example, in Canada, a relatively quiet country (although certainly not spared from TGT) over 50% of the population by the time they are about 40 years old will have had a mental disorder and the incidence is about 20% per year (let me find some references... Smetanin et al 2011 The life and economic impact of major mental illnesses in Canada). That then takes away credibility from the article.

The spelling of words that contained ‘s’ have been changed to show the ‘z’ where necessary.

Example cognisant changed to cognizant; recognise changed to recognized; etc. See track changes for all effected spelling changes.

The authors are of the opinion that a comprehensive case was made in section 2 (Materials and Methods) for the use of the constructivist paradigm. The aim of the study was to provide rich descriptions of people's lives and social worlds. As such individual perspectives were considered – relying as much as possible on the participants’ perceptions. The authors concede that perceptions are by nature not objective.

See line 1051…’the constructivist paradigm underpinning this paper assumes objectivity an impossibility’.

Line 25 has been changed to

‘South African society has an extensive history of race-based oppression’.

In addressing the limitation of the study, the authors acknowledge its strong regional/geographical focus (see line 1047).

Round 2

Reviewer 3 Report

Spelling is fine. The category I am forced to check and which I checked last time says: English language and style are fine/minor spell check required

I don't think the article is horrible to publish. No harm will come from psychologists reading it, i.e., it will probably have zero effect on patient treatment. However, it is unfortunate that the authors spend this effort interviewing people, writing text and so on, without having an underlying scientific basis to properly assess the information they have found. Maybe really the information they have found is useless, maybe really it is more important than they say it is. TGT is an important subject. So, it is unfortunate there is not better analysis of their hard work.
As I wrote last time, I am not rejecting the article, but trying to offer some insight into producing better work, in a more scientific framework. This article is fine for its purpose. However, if you wanted to design an electrical circuit or build a better suspension bridge or design a better psychiatric medication, you could not do so properly with the level of scientific rigor of studies such as this one -- the circuits will not work, the bridges will fall down, the psychiatric drugs will harm people.
I am not against the authors, I am not the enemy. I am someone who is simply saying you need to make better logical arguments based on better evidence. 
Although I have spent the last few decades providing help to persons with mental disorders, my original degree (i.e., undergraduate) was overwhelming based on the scientific method. I do not have the training (from one of the best universities in the world) to accept or understand statements the authors write in response to criticism such as, "See line 1051…’the constructivist paradigm underpinning this paper assumes objectivity an impossibility’.   If there is no objectivity, then again I don't have the background to evaluate it. And again, I am not against the subject matter. (By the way, as practicing psychologists and psychiatrists know, the field is filled to the brim with tools, scales, etc., in order to try to grasp some reality, some objectivity. In the USA, for example, it is medical malpractice not to use some sort of instrument to try to find some objectivity in mental health care.)
I wish the authors the best success. If the editor in charge is happy with the article, please go ahead and publish.